# Polymorphisms of the Vitamin D Binding Protein (VDBP) and Free Vitamin D in Patients with Cystic Fibrosis

**DOI:** 10.3390/nu16223850

**Published:** 2024-11-11

**Authors:** Paula Quesada-Colloto, Noelia Avello-Llano, Ruth García-Romero, María Garriga-García, Marina Álvarez-Beltrán, Ana Isabel Reyes-Domínguez, Ana Estefanía Fernández-Lorenzo, Helena Gil-Peña, Carlos Gómez-Alonso, Carmen García-Gil-Albert, Saioa Vicente-Santamaria, Luis Peña-Quintana, Juan José Díaz-Martin, José Ramon Gutiérrez-Martínez, Carmen Martin-Fernández, Agustín De la Mano-Hernández, Ana Moreno-Álvarez, David González-Jiménez

**Affiliations:** 1Primary Care Pediatric Service of Principality of Asturias, Centro de Salud de Ventanielles, 33010 Oviedo, Spain; 2Clinical Biochemistry Service, Hospital Universitario Central de Asturias (HUCA), 33011 Oviedo, Spain; 3Pediatric Gastroenterology and Nutrition Section, Hospital Universitario Miguel Servet, 50009 Zaragoza, Spain; 4Cystic Fibrosis Section, Hospital Universitario Ramón y Cajal, 28080 Madrid, Spain; 5Pediatric Gastroenterology and Nutrition Section, Hospital Universitario Vall d’Hebron, 08035 Barcelona, Spain; 6Pediatric Gastroenterology and Nutrition Section, Complejo Hospitalario Universitario Insular-Materno Infantil, 35016 Las Palmas de Gran Canaria, Spain; 7Pediatric Gastroenterology and Nutrition Section, Hospital Teresa Herrera, 15006 A Coruña, Spain; 8Department of Pediatrics, Hospital Universitario Central de Asturias (HUCA), Instituto de Investigación Sanitaria del Principado de Asturias (ISPA), 33011 Oviedo, Spain; 9Clinical Management Unit of Bone Metabolism, Hospital Universitario Central de Asturias (HUCA), 33011 Oviedo, Spain; 10Pediatric Gastroenterology and Nutrition Section, Hospital Ramon y Cajal, 28080 Madrid, Spain; 11Pediatric Gastroenterology and Nutrition Section, CIBER-OBM ISCIII, Complejo Hospitalario Universitario Insular-Materno Infantil, 35016 Las Palmas de Gran Canaria, Spain; 12Pediatric Gastroenterology and Nutrition Section, Hospital Universitario Central de Asturias (HUCA), 33011 Oviedo, Spain; 13Pediatric Pulmonology Section, Hospital Universitario Central de Asturias (HUCA), 33011 Oviedo, Spain; 14Pediatric Gastroenterology and Nutrition Section, Hospital Infantil Universitario Niño Jesús, 28009 Madrid, Spain; 15Pediatric Gastroenterology and Nutrition Section, Complejo Hospitalario Universitario A Coruña, 15006 A Coruña, Spain

**Keywords:** vitamin D, cystic fibrosis, VDBP, free vitamin D, free vitamin D hypothesis

## Abstract

Objectives/Background: Vitamin D-binding protein (VDBP) and free vitamin D are new markers that are being studied as a possible markers of vitamin D status. The main aim of our study was to analyze the VDBP genotype and quantify the levels of free vitamin D in a sample of cystic fibrosis (CF) patients. Methods: We conducted a multicenter, cross-sectional, and prospective study including patients with CF and exocrine pancreatic insufficiency who were clinically stable. We investigated vitamin D levels (total and free) and the different VDBP haplotypes. Free vitamin D levels were measured using an electro-chemiluminescence assay. Results: A sample of 48 patients was obtained (52% male; median age 13.8 years). The most common allele of VDBP was Gc1s (72%) > Gc2 (52%) > Gc1f (27%). The median calcidiol was 21.2 ng/mL (IR 15.3–26.9), and 81% had levels in the insufficiency range: 23 patients (48%) below 20 ng/mL, and 16 (33%) between 20 and 30 ng/mL. The median free vitamin D level was 4.2 pg/mL (IR 3.9–5.6). A positive correlation was observed between calcidiol and free vitamin D levels (r = 0.871; *p* < 0.0001). After adjustment for season, vitamin D supplementation, sex, and CF-related diabetes, patients with Gc1f polymorphism had a lower risk of vitamin D deficiency, OR 0.22 (95% CI 0.05–0.99), and *p* = 0.027. A negative linear trend was observed between the polymorphisms grouped into three categories (Gc1/Gc1, Gc1/Gc2, and Gc2/Gc2, in that order) and vitamin D and free vitamin D levels (*p* = 0.025 and *p* = 0.033, respectively). Conclusion: In CF, as in the general population, the most common VDBP haplotype in the Caucasian race is Gc1s. VDBP polymorphisms influence serum vitamin D and free vitamin D levels in CF patients. There is a good correlation between free vitamin D and calcidiol levels, suggesting that measuring the latter in CF does not seem to provide any additional benefit.

## 1. Introduction

Vitamin D deficiency has traditionally been associated with bone health disorders. In recent years, however, it has received attention due to the association between low levels of this vitamin and cardiovascular risk, as well as inflammatory, infectious, and neoplastic processes [1].

Current guidelines recommend that vitamin D status be monitored by measuring 25 OH vitamin D (calcidiol) levels. While calcidiol offers several advantages, including a long half-life (2–3 weeks) and greater stability, there are other considerations: on the one hand, there is significant methodological variability in its measurement; on the other hand, the association between vitamin D status and bone health markers, as well as other extra-skeletal functions, is not particularly strong in pediatric populations and is subject to variation according to specific factors, including age, race, obesity, medication use, genetic factors, and chronic diseases. For these reasons, efforts have been made in recent years to advance the understanding of new markers, including free vitamin D.

Vitamin D and its metabolites are closely related to serum proteins, with the vitamin D-binding protein (VDBP) representing the most significant of these (accounting for 85–90% of the total). Approximately 10–15% is bound to albumin, with less than 0.1% circulating freely [2,3]. VDBP is a protein produced by the liver and encoded by the GC gene (location: 4q13.3). There are several known single-nucleotide polymorphisms (SNPs) in the GC gene, with rs7041 (p.Asp416Glu) and rs4588 (p.Thr420Lys) being the most investigated. Due to the allelic combination of these SNPs, three main haplotypes have been described (Gc1F, Gc1S, and Gc2), and they frequency vary according to race and geographical distribution. It is well established that these haplotypes determine the concentration of VDBP, its affinity for 25 OH vitamin D, and, consequently, the levels of free vitamin D [4,5,6]. These findings could explain the lack of correlation between the total 25 OH vitamin D levels and clinical manifestations [7].

Additionally, the role of VDBP haplotypes is under study for their potential impact on diseases beyond the skeletal system. Studies conducted in recent years suggest that, depending on the presence of certain haplotypes, the risk of some diseases could increase, including oncological (breast, prostate, and colorectal cancer), rheumatological (rheumatoid arthritis and spondylitis), endocrinological (diabetes mellitus, obesity, and autoimmune thyroid diseases), pulmonary (chronic obstructive pulmonary disease, asthma, and tuberculosis), and cardiovascular diseases (coronary artery disease) [8,9].

Furthermore, VDBP and albumin levels can vary in patients with certain conditions. For instance, they tend to decrease in cases of cirrhosis or in elderly patients with multiple comorbidities. Conversely, they increase in the last months of pregnancy. In such circumstances, both calcidiol and free vitamin D levels may fluctuate, prompting the suggestion that the latter may more accurately reflect vitamin D status [10,11,12].

Given that cystic fibrosis (CF) is a disease characterized by a state of chronic inflammation and alterations in oxidative stress alterations, with pulmonary exacerbations, poor intake, and pancreatic insufficiency often resulting in hypoproteinemia and malnutrition, it could be hypothesized that the determination of free vitamin D may be of particular interest in this condition.

The main objective of our study was to analyze the genotype of vitamin D-binding protein (VDBP) and free vitamin D in a sample of clinically stable cystic-fibrosis patients.

## 2. Materials and Methods

### 2.1. Study

This is a multicenter, cross-sectional, and prospective study involving patients with CF and exocrine pancreatic insufficiency who are clinically stable. We included patients from 4 different hospitals from Spain. Each center contributed between 6 and 23 patients. The study examined levels of vitamin D (total and free), as well as the VDBP different haplotypes. Approval for the study was granted by the clinical research ethics committee of Hospital Universitario Central de Asturias (HUCA) (number 67/19), and the committees of the rest of the participating hospitals also gave their approval. All participants between 12 and 18 years gave their consent to participate in the study, and adult patients and parents of minors signed informed-consent forms to participate in the study.

### 2.2. Subjects

#### 2.2.1. Inclusion Criteria

Patients diagnosed with CF in a clinically stable condition, defined by the following clinical criteria: absence of cough, fever, expectoration, or hemoptysis, and must not have undergone any oral or intravenous antibiotic treatment in the 2 weeks prior to their inclusion in the study.

CF patients with exocrine pancreatic insufficiency defined by a fecal elastase determination < 200 mcg/g of stool.

#### 2.2.2. Exclusion Criteria

Platelet count < 50,000/mm^3^.

Significant liver function impairment, severe cholestasis, or renal insufficiency.

Hospitalization or administration of oral or intravenous antibiotics within the 2 weeks prior to the start of the study.

### 2.3. Study Development and Response Evaluation

All analytical determinations were collected at hospitals of origin, stored at appropriate temperatures, and subsequently analyzed in a centralized manner.

#### 2.3.1. Genotyping of the Vitamin D-Binding Protein (VDBP)

Blood samples were collected in 9 mL EDTA tubes and frozen at −20 °C until shipment. Upon receipt at the destination laboratory, the blood, once thawed, was subjected to the salt-based genomic DNA-extraction method. The areas of the GC gene (encoding VDBP) that contain the SNPs rs7041 and rs4588 were amplified by polymerase chain reaction (PCR) using the specific primers: 5′ TGTAAAAGATCTGAAATGG 3′ and 3′ CATAATGGCATCTCAATAA 5′ (annealing of 48°), following the thermocycling protocol used by Cillero et al. [7].

Amplicons were purified and sequenced by automated Sanger sequencing using the same primers individually. The reading of the electropherograms for the detection of SNPs was carried out with the Sequencher 5.4 software (Gene Codes Corporation) (Table 1).

#### 2.3.2. Total Vitamin D Levels (25OH Vitamin D, Calcidiol)

Serum total 25OHD concentration was measured by an electro-chemiluminescence assay (Roche Diagnostics GmbH, Mannheim, Germany). For serum 25OHD, the inter-assay coefficient of variation (CV) determined with PreciControl VitDT 1 and 2 ranged from 3.76% to 10.2%.

Vitamin D (calcidiol) levels were considered deficient when below 20 ng/mL, insufficient between 20 and 30 ng/mL, and sufficient when above 30 ng/mL [11].

#### 2.3.3. Free Vitamin D Levels

Direct measurement of free 25OHD concentration was performed by a competitive ELISA assay (KARF1991, DiaSource, Louvain-la-Nueve, Belgium), which detects the free fraction. The analytical sensitivity obtained ranged from 2.4 to 3.9 pg/mL.

#### 2.3.4. Identification and Epidemiological Data

Identification by codes assigned to the hospital and the patient, date of birth, sex, clinical form, and genetics.

#### 2.3.5. Concomitant Medication

Pancreatic enzymes (lipase UI/kg/day), corticosteroids (oral or inhaled), CFTR potentiators and modulators (Lumacaftor/Ivacaftor (Orkambi^®^), Tezacaftor/Ivacaftor (Symkevi^®^)), and vitamin D (UI/day).

#### 2.3.6. Anthropometry

The weight and height of each patient were directly obtained, with the patient barefoot and in underwear, using instruments with a precision of 50 g and 0.5 cm, respectively. Calculation of anthropometric parameters was performed using the nutritional tool of the Spanish Society of Pediatric Gastroenterology, Hepatology and Nutrition (SEGHNP; www.seghnp.org, accessed on 1 October 2023). Body mass index (BMI) was calculated and expressed by percentile and z-score using the World Health Organization 2006 (WHO; https://www.who.int, accessed on 1 October 2023) reference values for children under 6 years of age and Carrascosa 2010 reference values [13] for children over 6 years. The nutritional status of each patient was classified according to the criteria agreed upon by the North American and European CF societies (based on BMI in adults and percentiled BMI in children) as follows [14]: undernourished (<18.5 kg/m^2^; <P10), at nutritional risk (18.5–21.9 kg/m^2^; P10–P49), well-nourished (22–24.9 kg/m^2^; P50–P84), overweight (25–29.9 kg/m^2^, P85–P94), and obese (≥30 kg/m^2^; ≥P95).

### 2.4. Sample Size

To calculate the sample size, we estimated the mean free vitamin D. Based on the recent literature [15], in a sample of clinically stable cystic-fibrosis patients, the mean was 5.9 pg/mL; SD = 1. Assuming an alpha error of 0.05 and a beta of 0.10, the sample size needed to estimate the mean with an absolute error of 0.5 pg/mL was 43 patients.

### 2.5. Statistical Analysis

Study data were collected and managed using REDCap [16] electronic data-capture tools hosted at SEGHNP (www.seghnp.org, accessed on 1 October 2023). Technical support was provided by the AEG REDCap Support Unit, shared with the Spanish Association of Gastroenterology (AEG). REDCap (Research Electronic Data Capture) is a secure web-based application designed to support data capture for research studies, providing an intuitive interface for validated data entry, audit trails for tracking data manipulation and export procedures, automated export procedures for seamless data downloads to common statistical packages, and procedures for importing data from external sources. Statistical analysis was performed using the Statistical Package (STATA), version 18.0.

Basic statistical techniques of descriptive analysis were used in the study. Kolmogorov–Smirnov tests were used to assess normality. Pearson and Spearman correlation tests were used to analyze the joint behavior of quantitative variables. Two-tailed *t*-tests were used to compare means between 2 groups. Chi-square tests were used for comparison of proportions. Nonparametric tests (Kruskal–Wallis for comparison of means between three or more groups) were used for parameters that did not meet statistical normality criteria. Differences were considered statistically significant when *p*-values were less than 0.05.

## 3. Results

### 3.1. Sample Description

A sample of 48 patients was obtained. In total, 52% of the sample was male. The median age was 13.75 years (range 6–46). Only one-third of the sample had been diagnosed with CF through neonatal screening. All patients had pancreatic insufficiency, and only 11% had a history of meconium ileus. In total, 56.25% were homozygous for the delta F508 mutation, 39.6% were heterozygous, and only two patients (4.15%) were non-carriers of this mutation. A total of 28% of patients were receiving corticosteroid treatment, and 31% were on CFTR modulators. Moreover, 23% had liver involvement without cirrhosis, and 21% had glucose metabolism disorders.

The characteristics of the sample are summarized in Table 2. Patients’ concomitant medication is shown in Table 3:

### 3.2. Haplotype Description

The most common allele was Gc1s, present in 72% of patients, followed by Gc2, present in 52%, and the least common allele was Gc1F, present in 27%. Regarding the combination of haplotypes, the most frequent was Gc1/Gc1 (48%), followed by Gc1/Gc2 (39.5%) and Gc2/Gc2 (12.5%). Considering the six possible haplotype combinations, the most frequent was Gc1s/Gc2, present in one-third of the sample (33%), followed by Gc1s/Gc1s (27%), Gc1s/Gc1f (12.5%), Gc2/Gc2 (12.5%), Gc1f/Gc1f (8.5%), and Gc1f/Gc2 (6.5%).

### 3.3. Description of Vitamin D Levels and Dosages

All patients received vitamin D supplementation. Doses ranged between 500 and 7000 IU per day (mean, 2646 IU; SD, 1602). The median total vitamin D (calcidiol) level of the patients included in the sample was 21.2 ng/mL (interquartile range of 15.25–26.85). Nine patients (18.7%) had levels within the sufficiency range. The remaining 81.3% had levels within the insufficiency range: 23 patients (48%) had levels below 20 ng/mL (deficiency), and 16 (33.3%) had levels between 20 and 30 ng/mL. The median free vitamin D level was 4.245 pg/mL (interquartile range, 3.9–5.57). Twenty-one patients in the sample (43.75%) had levels below 3.9 pg/mL. A positive correlation was observed between calcidiol and free vitamin D levels (r = 0.8709; *p* < 0.0001). There were other correlations observed, but none of them was statistically significant: calcidiol with vitamin D dose (r = 0.1242; *p* = 0.4005), calcidiol with age (r = 0.1078; *p* = 0.4659), free vitamin D with vitamin D dose (r = 0.1206; *p* = 0.4141), and free vitamin D with age (r = 0.1473; *p* = 0.3178).

### 3.4. Description of Calcidiol and Free Vitamin D Levels According to Haplotypes (Table 4, Table 5, Table 6 and Table 7)

#### 3.4.1. Vitamin D and Free Vitamin D Levels According to the Gc1f-Gc1s or Gc2 Haplotype (Table 4)

Patients with the Gc2 polymorphism exhibited a higher proportion of total vitamin D levels within the insufficiency range: 92% vs. 69.5% in patients without this polymorphism (*p* = 0.047). After adjusting for potential confounding factors, including season, vitamin D supplement dosage, sex, and CF-related diabetes diagnosis, the OR was determined to be 13.46 (95% CI, 1.35–133; *p* = 0.027).

**Table 4 nutrients-16-03850-t004:** Comparison of total and free vitamin D levels according to the presence of the 3 existing haplotypes.

	Gc1f			Gc1s			Gc2		
	Yes (n = 13)	No (n = 35)	*p*	Yes (n = 35)	No (n = 13)	*p*	Yes (n = 25)	No (n = 23)	*p*
Total vitamin D (ng/mL)	25.50 ± 10.30	20.50 ± 8.60	0.095	22.10 ± 8.90	21.30 ± 10.40	0.776	19.50 ± 7.40	24.40 ± 10.50	0.063
Free vitamin D (pg/mL)	5.40 ± 2.40	4.40 ± 1.50	0.124	4.70 ± 1.60	4.80 ± 2.40	0.846	4.20 ± 1.50	5.30 ± 2.00	0.033

Data are expressed as mean and standard deviation (*t*-test for independent samples).

In contrast, in carriers of the Gc1f polymorphism, a lower proportion of patients with deficient vitamin D levels was observed, 23% vs. 57% (*p* = 0.036). After adjusting for potential confounding factors, including season, vitamin D supplement dosage, sex, and CF-related diabetes diagnosis, the OR was determined to be 0.22 (95% CI, 0.05–0.99, *p* = 0.027). A lower proportion of patients with free vitamin D levels below 3.9 ng/mL was also observed in carriers of this haplotype: 23% vs. 54.3% in patients without this polymorphism (*p* = 0.054). After adjusting for the aforementioned factors, the OR was 0.25 (95% CI, 0.06–1.11; *p* = 0.070) (Table 5).

**Table 5 nutrients-16-03850-t005:** Prevalence of vitamin D insufficiency and deficiency according to the presence or absence of Gc2 and Gc1f.

	Gc2			Gc1f		
	Yes (n = 25)	No (n = 23)	*p*	Yes (n = 13)	No (n = 35)	*p*
25 oh vitamin D < 20 ng/mL n (%)	14 (56%)	9 (39%)	0.243 *	3 (23%)	20 (57%)	0.036 *
25 oh vitamin D < 30 ng/mL n (%)	23 (92%)	16 (70%)	0.047 *	9 (69%)	30 (86%)	0.194 *

* Ch2 test.

#### 3.4.2. Vitamin D and Free Vitamin D Levels According to the Haplotypes Divided into 3 Categories (Gc1/Gc1, Gc1/Gc2, and Gc2/Gc2) (Table 6)

The subgroup analysis revealed statistically significant differences between carriers of the Gc1/Gc1 haplotypes and Gc2/Gc2 carriers (*p* = 0.036) and between Gc1/Gc2 carriers and Gc2/Gc2 carriers (*p* = 0.048).

**Table 6 nutrients-16-03850-t006:** Vitamin D and free vitamin D levels according to haplotypes divided into 3 categories.

Haplotypes	Vitamin D	*p*	Free Vitamin D	*p*
Gc1/Gc1	22.80 (16.20–32.30) *	0.059	4.74 (3.90–6.68)	0.539
Gc1/Gc2	21.00 (15.40–24.60) *		4.14 (2.82–5.34)	
Gc2/Gc2	14.5 (11.30–16.80)		3.9 (2.57–3.90)	

* vs. Gc2/Gc2 (*p* < 0.050). Data are expressed as median and interquartile range (Kruskal–Wallis).

#### 3.4.3. Vitamin D and Free Vitamin D Levels According to the Haplotypes Divided into 6 Categories (Gc1f/Gc1f, Gc1f/Gc1s, Gc1f/Gc2, Gc1s/Gc1s, and Gc2/Gc2) (Table 7)

The subgroup analysis revealed statistically significant differences between carriers of the Gc1f/Gc2 haplotypes and Gc2/Gc2 carriers (*p* < 0.050).

**Table 7 nutrients-16-03850-t007:** Vitamin D and free vitamin D levels according to haplotypes divided into 6 categories. Data are expressed as median and interquartile range (Kruskal–Wallis).

Haplotypes	Vitamin D	*p*	Free Vitamin D	*p*
Gc1f/Gc1f	30.70 (16.65–41.70)	0.263	6.63 (3.57–9.41)	0.368
Gc1f/Gc1s	21.55 (18.10–32.30)		4.78 (3.90–6.76)	
Gc1f/Gc2	23.2 (21.00–24.60) *		4.65 (4.14–4.94)	
Gc1s/Gc1s	23.90 (16.20–29.80)		4.67 (3.90–5.95)	
Gc1s/Gc2	17.25 (15.35–25.20)		3.98 (2.75–5.44)	
Gc2/Gc2	14.55 (11.30–16.80)		3.90 (2.57–3.90)	

* vs. Gc2/Gc2 (*p* < 0.050).

#### 3.4.4. Trend Between Polymorphisms Grouped into 3 Categories (Figure 1 and Figure 2)

A Jonckheere–Terpstra test showed that there was a statistically significant trend of higher median calcidiol and free vitamin D with VDBP polymorphisms ordered by VDBP affinity (from Gc2/Gc2 and Gc1/Gc2 to Gc1/Gc1), T_JT_ = 223, z = 2.189, and *p* = 0.026; and T_JT_ = 215, z = 2.129, and *p* = 0.032 respectively.

**Figure 1 nutrients-16-03850-f001:**
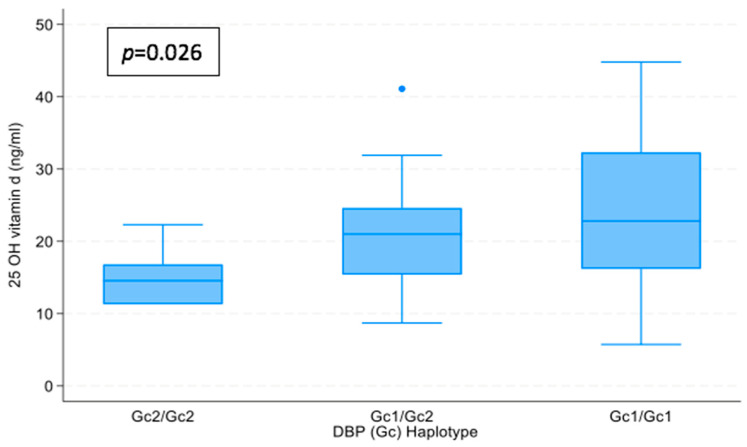
Relationship between polymorphisms grouped into 3 categories ordered by VDBP affinity (from Gc2/Gc2 and Gc1/Gc2 to Gc1/Gc1) and 25 OH Vitamin D levels. *p* = nonparametric tests for trend: Jonckheere–Terpstra test.

**Figure 2 nutrients-16-03850-f002:**
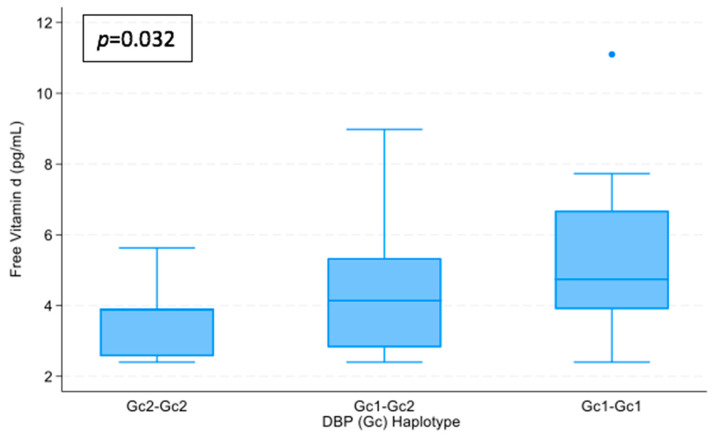
Relationship between polymorphisms grouped into 3 categories ordered by VDBP affinity (from Gc2/Gc2 and Gc1/Gc2 to Gc1/Gc1) and Free vitamin D levels. *p* = nonparametric tests for trend: Jonckheere–Terpstra test.

## 4. Discussion

In the present study, we have analyzed the different VDBP haplotypes, as well as total and free vitamin D levels, in a sample of pediatric and adult patients with stable cystic fibrosis. Regarding calcidiol levels, it was observed that only a minority of patients exhibited values within the sufficiency range. This trend has already been identified in previous studies conducted in patients with CF [15,17,18]. This may have implications not only for bone health but also for the anti-inflammatory and antibacterial roles, among others, of vitamin D, which in turn can significantly impact the clinical evolution and stability of patients with CF, given the characteristics of their disease [18].

For several years now, there has been an effort to advance the understanding of new markers for assessing the nutritional status of vitamin D. The free vitamin D theory suggests that the free portion, unbound to proteins, is biologically active and could more accurately reflect nutritional status. In the present study, a robust linear correlation was observed between vitamin D and free vitamin D levels, a circumstance previously described by Lee et al. [17] and subsequently replicated in both healthy populations and patients without hypoalbuminemia or hypoproteinemia [19,20]. Historically, CF has been associated with an elevated risk of malnutrition. However, since the implementation of neonatal screening, the use of high-calorie diets, management in accredited units, and the introduction of CFTR modulators, there has been a progressive reduction in malnutrition and an improvement in albumin levels. In both the study conducted by Lee et al. [17] and our own, over 80% of patients were well-nourished. This circumstance could partly explain the good correlation between the two markers. Our findings do not indicate that determining free vitamin D in patients with CF provides more valuable information or offers greater benefit in assessing vitamin D nutritional status compared to calcidiol.

The most prevalent VDBP haplotype was Gc1s, followed by Gc2 and, subsequently, Gc1f. While specific racial data were not available, the origin of the samples allows for the reasonable assumption that a majority of the patients were Caucasian. Our findings are consistent with those of previous studies investigating other pathologies [21] and in healthy populations, where the most frequent haplotype in the Caucasian race is Gc1s, while the Gc1f haplotype is more frequent in the black and Asian races [4]. Regarding haplotype combinations, the most frequent combination observed in other studies in the Caucasian race was Gc1s/Gc2 [22], which is consistent with the results of the present study. The least-frequent combination observed in other studies was Gc1F/Gc1F [7], whereas in the present study, it was Gc1F/Gc2.

The relationship between VDBP polymorphisms and total and free vitamin D levels was investigated. VDBP polymorphisms determine the degree of affinity of vitamin D for VDBP, as well as changes in calcidiol levels. In our study, higher levels of both vitamin D and free vitamin D were observed in those with greater affinity GC1-1 > GC1-2 > GC2-2, while those with at least one GC2 polymorphism were at higher risk of vitamin D deficiency, regardless of other factors. To the best of our knowledge, no studies on VDBP polymorphisms in CF patients have been conducted. Consequently, we are unable to make direct comparisons with studies in healthy populations and patients with other pathologies. In this regard, our data are consistent with the findings of the review by Bouillon et al. [23], which describes a 5–15% reduction in calcidiol levels associated with the Gc2-2 haplotype, as observed in several studies conducted on diverse populations [24,25,26]. Furthermore, patients carrying the Gc1f allele exhibited a reduced risk of vitamin D deficiency. Nevertheless, other studies, such as that conducted by Schwartz et al. [21], did not identify a clear linear correlation between the variables under investigation, although differences between haplotypes were observed.

The results obtained reinforce the significance of genetic factors in elucidating the mechanisms of vitamin D metabolism [27]. Our data suggest that the Gc1f polymorphism would be the most “favorable” for vitamin D and free vitamin D levels, particularly when present alongside another Gc1f or -s polymorphism. It is noteworthy that this polymorphism is precisely the least frequent in the Caucasian race, while Gc2 was most related to deficiency.

Recently, through whole-genome sequencing (WGS), various Polygenic Risk Scores (PRSs) have been developed, which play a pivotal role in the different responses to vitamin D supplementation observed in CF patients. In the study by Lai, Song, et al. [28], in which CF patients diagnosed through neonatal screening were studied and followed up during the first 3 years of life, a subset of the sample exhibited insufficient vitamin D levels despite receiving adequate supplementation doses. Whole-genome sequencing (WGS) was performed, and a PRS was established based on the observed polymorphisms. In the multivariate analysis, PRS was found to be the most important factor related to nutritional status. Conversely, polymorphisms in other genes, including GC, LIPC, CYP24A1, and PDE3B, were associated with patient responsiveness to supplementation. In another study with a similar methodology (WGS with subsequent PRS determination) conducted by Braun et al. [29] on a sample of 80 adult CF patients receiving adequate vitamin D supplementation, a significant correlation between PRS and total vitamin D levels was also observed, thereby providing further support for this theory.

It is important to note that, at the time of the study, the triple therapy or high-efficacy therapy comprising the combination of the three modulators of the cystic fibrosis transmembrane conductance regulator (CFTR) channel, Elexacaftor–Tezacaftor–Ivacaftor, had not yet been approved. Consequently, none of the patients included was receiving this medication and its associated benefits [30].

It is important to highlight the key strengths of this study. Its first aim was to analyze VDBP polymorphisms in CF patients and relate them to total and free vitamin D levels. It is also a multicenter study with determinations centralized and performed using the same methodology, thereby ensuring the reliability of the results. The calculation of free vitamin D using formulas, as has been traditionally performed, does not appear to be an accurate approach. The determination of free vitamin D using the ELISA technique, which was used in this work, is more accurate than determining it through formulas. There are articles that compare both methods, and the results are different [11].

As limitations, it should be noted that although vitamin D supplementation of the included patients is based on common criteria, being a multicenter study, the patients belong to different geographical areas within the same country, with varying sun exposure. This could affect total calcidiol and free vitamin D levels, which may in turn impact the results of the study. However, although it was not directly measured, the season of the year when the calcidiol determination was performed was considered in the analysis of the results, so it is indirectly analyzed as well. In the present study, patient adherence to prescribed vitamin D supplementation was not directly studied. Nevertheless, CF patients usually receive vitamin D supplementation through multivitamin complexes that also include vitamins A (retinol) and E (alfa-tocoferol), which were measured, showing normal values in all patients except in one of them, suggesting that adherence to supplementation in almost all the patients included in the study was adequate.

## 5. Conclusions

In CF, as in the general population, the most frequent VDBP polymorphism in the Caucasian race is Gc1s. VDBP polymorphisms influence serum vitamin D and free vitamin D levels in CF patients. There is a good correlation between free vitamin D and calcidiol levels, indicating that the determination of the latter in CF does not seem to provide additional benefits. VDBP polymorphisms influence serum vitamin D and free vitamin D levels in CF patients.

## Figures and Tables

**Table 1 nutrients-16-03850-t001:** Correlation between SNPs rs7041 and rs4588 and kind of *GC* gene allele.

SNP rs7041 NM_000583.4 (*GC*): c.1296T>G; p.D432E		SNP rs4588 NM_000583.4 (*GC*): c.1307C>A; p.T436K		SNP-Based Allele
Nucleotide	Amino Acid	Nucleotide	Amino Acid	
GAT	D	ACG	T	1F
GAG	E	ACG	T	1S
GAT	E	AAG	K	2

**Table 2 nutrients-16-03850-t002:** Demographic, genetic, and clinical characteristics of the patients.

Item		Value
Sex (n male, (%))		25 (52%)
Age (median, interquartile range)		13.75 (10.5–20.0)
Months at diagnosis (median, interquartile range)		27.48 (2–48)
Diagnosis by neonatal screening (n, %)		16 (33%)
History of meconium ileus (n, %)		5 (11%)
Exocrine pancreatic insufficiency (n, %)		48 (100%)
Genetics (n, %)	Homozygous df508	27 (56.2%)
	Heterozygous df508	19 (39.6%)
	Non-carriers of df508	2 (4.2%)
Liver disease associated with CF (liver involvement without cirrhosis (n, %)		11 (23%)
Glucose metabolism disorder (n, %)		10 (21%)
Vitamin A deficiency (serum retinol < 20 mcg/dL)		1 (2%)
Vitamin E deficiency (serum alfa-tocoferol < 500 mcg/dL)		1 (2%)
*Pseudomonas aeruginosa* colonization (n, %)		8 (17%)
Baseline nutritional status (n, %)	Malnutrition	7 (17%)
	Normal nutrition	34 (81%)
	Overweight	1 (2%)
Nutritional goal	(BMI > p50) (n, %)	27 (64%)

**Table 3 nutrients-16-03850-t003:** Patients’ concomitant medication.

Medication	Value
Pancreatic enzymes (n, %)	48 (100%)
Dose (UI/kg/day) (mean, interquartile range)	5363 (3673–6926)
Vitamin D supplements (n, %)	48 (100%)
Dose (UI/day) (mean, standard deviation)	2646 (1602)
Corticosteroids (n, %)	13 (28.0%)
Oral corticosteroids (n, %)	2 (15.4%)
Inhaled corticosteroids (n, %)	11 (84.6%)
CFTR modulators/potentiators (n, %)	14 (31.0%)
Lumacaftor/Ivacaftor (Orkambi^®^) (n, %)	4 (28.5%)
Tezacaftor/Ivacaftor (Symkevi^®^) (n, %)	10 (71.5%)

## Data Availability

The dataset originated from the trial is available upon request to the corresponding author.

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
