# Peer review of "Polymorphisms of the Vitamin D Binding Protein (VDBP) and Free Vitamin D in Patients with Cystic Fibrosis"

_nutrients, 2024, doi:10.3390/nu16223850_

Round 1

Reviewer 1 Report

Comments and Suggestions for Authors

Review of the manuscript titled:

Polymorphisms of the Vitamin D Binding Protein (VDBP) and Free vitamin D in pediatric and young adult patients with cystic fibrosis.

Authors: Paula Quesada-Colloto, Noelia Avello-Llano, Ruth García-Romero,María Garriga-García, Marina Álvarez-Beltrán, Ana IsabelReyes-Domínguez, Ana Estefanía Fernández-Lorenzo, Helena Gil-Peña, CarlosGómez-Alonso, Carmen García-Gil-Albert, Saioa Vicente-Santamaria, LuisPeña-Quintana, Juan José Díaz-Martin, José Ramon Gutiérrez-Martínez,Carmen Martin-Fernández, Agustín De la Mano-Hernández, AnaMoreno-Álvarez, David González-Jiménez

The manuscript presents a study investigating the polymorphism of vitamin D binding protein (VDBP), total and free vitamin D concentrations in cystic fibrosis (CF) patients with exocrine pancreatic insufficiency aged 6 to 46 years.

The Authors used automated Sanger sequencing for VDBP genotyping, electrochemiluminescence assay and commercially available competitive ELISA kit to estimate total vitamin D (25 OH vitamin D - calcidiol) and free vitamin D concentrations.

For the first time, the study assesses the VDBP polymorphism related to total and free vitamin D levels in patients with CF, which is a strength of the study. However, the results confirmed what has been documented in patients without CF. The weakness of the study, apart from those mentioned by the Authors, is the small sample size (especially for the multicenter study including pancreatic insufficiency patients).

Comments and suggestions for Authors:

1. The current title of the manuscript refers only to the assessment of the VDBP polymorphism and free vitamin D. Total vitamin D levels were also determined in this study. In addition, the age range of the study group was 6 to 46 years. This age range doesn't only apply to children and young people. The title of the article should be corrected, e.g. "Polymorphisms of the vitamin D binding protein (VDBP) and vitamin D in paediatric and adult patients with cystic fibrosis" or "Polymorphisms of the vitamin D binding protein (VDBP) and vitamin D status in pediatric and adult patients with cystic fibrosis"?

2. In the Abstract section (lines 46-48), it is written that: "The main aim of our study was to analyse the VDBP genotype and to quantify the levels of VDBP and free vitamin D in a sample of cystic fibrosis (CF) patients". However, the study did not investigate the quantitative level of VDBP, only its polymorphism. The same sentence appears in the Introduction, line 102.

3. Were the CF patients receiving other medications besides probiotics, corticosteroids, potentiators, modulators, and pancreatic enzymes? The sentence referring to this aspect ends with "..." (Material and Methods section, lines 154-155).

4. How was the sample size calculated?

5.  The Authors documented the results of the ch2 test (the Results section, lines 217-219 and 225-226). How was the chi2 test calculated? Tables showing the prevalence of vitamin D insufficiency or its deficiency according to the presence or absence of Gc2 and Gc1f would be helpful.

6. In Table 3, the Authors reported that 25 CF patients had the Gc2 haplotype, while 28 did not. The study group consisted of 48 CF patients, not 53 subjects. In addition, in Table 3, the Authors probably used the word „Si” instead of „Yes”.

7. Standardization of decimals for p-value in tables and figures would be advisable.

8. Table 5 and Figures 1 and 2 refer to the same results. It may be sufficient to show either a table or figures. Moreover, Table 5 shows statistical significance between genotypes Gc1f/Gc2 and Gc2/Gc2, while Figure 1 shows statistical significance between Gc1s/Gc2 and Gc2/Gc2. In Figure 1, the p-value of 0.0536, which is close to statistical significance, is not marked with two asterisks - information below Figure 1. A small note in the heading of Table 5: The Authors probably entered "free vitamin D libre" instead of "free vitamin D level".

9. In the Discussion section, line 276, the word "young" could be removed if the Authors decide to change the title of the manuscript.

10.  Standardization of the References section is necessary.

Author Response

Comments 1: The current title of the manuscript refers only to the assessment of the VDBP polymorphism and free vitamin D. Total vitamin D levels were also determined in this study. In addition, the age range of the study group was 6 to 46 years. This age range doesn't only apply to children and young people. The title of the article should be corrected, e.g. "Polymorphisms of the vitamin D binding protein (VDBP) and vitamin D in paediatric and adult patients with cystic fibrosis" or "Polymorphisms of the vitamin D binding protein (VDBP) and vitamin D status in pediatric and adult patients with cystic fibrosis"?

Response 1: We totally agree with this comment. Therefore, we change it to: “Polymorphisms of the Vitamin D Binding Protein (VDBP) and Free vitamin D in patients with cystic fibrosis”. We erase “young adult” and we don’t specify the age of the patients included in the study in the tittle (Page 1, lines 1-3).

Comments 2: In the Abstract section (lines 46-48), it is written that: "The main aim of our study was to analyse the VDBP genotype and to quantify the levels of VDBP and free vitamin D in a sample of cystic fibrosis (CF) patients". However, the study did not investigate the quantitative level of VDBP, only its polymorphism. The same sentence appears in the Introduction, line 102. 

Response 2: Thank you for pointing this out. We didn’t quantify the levels of VDBP, so it’s a mistake and we have already erased it.

Comments 3: Were the CF patients receiving other medications besides probiotics, corticosteroids, potentiators, modulators, and pancreatic enzymes? The sentence referring to this aspect ends with "..." (Material and Methods section, lines 154-155).

Response 3: It is true that we had left the section incomplete. We have modified it and specified the medications, as well as the type and quantity of those included in the study: 2.3.5. Concomitant medication: Pancreatic enzimes (lipase UI/kg/day), corticosteroids (oral or inhalated), CFTR potentiators and modulators (Lumacaftor/Ivacaftor (Orkambi), Tezacaftor/Ivacaftor (Symkevi)) and vitamin D (UI/day) (Page 4, lines 166-168).

Comments 4: How was the sample size calculated? 

Response 4: To calculate the sample size, we estimated the mean free vitamin D. Based on recent literature 1, in a sample of clinically stable cystic fibrosis patients, the mean was 5.9 pg/mL; SD = 1. Assuming an alpha error of 0.05 and a beta of 0.10, the sample size needed to estimate the mean with an absolute error of 0.5 pg/mL was 43 patients. We added this explanation to the methods section (page 4, lines 184-188).

[1] Lee MJ, Kearns MD, Smith EM, Hao L, Ziegler TR, Alvarez JA, Tangpricha V. Free 25-Hydroxyvitamin D Concentrations in Cystic Fibrosis. Am J Med Sci. 2015 Nov;350(5):374-9. doi: 10.1097/MAJ.0000000000000592

Comments 5: The Authors documented the results of the ch2 test (the Results section, lines 217-219 and 225-226). How was the chi2 test calculated? Tables showing the prevalence of vitamin D insufficiency or its deficiency according to the presence or absence of Gc2 and Gc1f would be helpful. 

Response 5: Agree. We have, accordingly, done this table to show the prevalence of vitamin D insufficiency and deficiency according to the presence or absence of Gc2 and Gc1f, so it would be easier to understand. We added it to the results section (pages 7, lines 251-253).

Table 5.  Prevalence of vitamin D insufficiency and deficiency according to the presence or absence of Gc2 and Gc1f

Gc2

Gc1f

Yes 

(n=25)

No

(n=23)

p

Yes

(n=13)

No

(n=35)

p

25 oh vitamin D <20 ng/ml n (%)

14 (56%)

9 (39%)

0.243*

3 (23%)

20 (57%)

0.036*

25 oh vitamin D < 30 ng/ml n (%)

23 (92%)

16 (70%)

0.047*

9 (69%)

30 (86%)

0.194*

* Ch2 test

Comments 6: In Table 3, the Authors reported that 25 CF patients had the Gc2 haplotype, while 28 did not. The study group consisted of 48 CF patients, not 53 subjects. In addition, in Table 3, the Authors probably used the word „Si” instead of „Yes”.

Response 6: Thank you for pointing that. The numbers of patients who did not have Gc2 haplotype was 23 instead of 28. We have changed it (pages 6-7, lines 247-249):

Gc1f

Gc1s

Gc2

Yes

(n-=13)

No

(n-=35)

p

Yes

(n-=35)

No

(n-=13)

P

Yes

(n-=25)

No

(n-=23)

P

Total vitamin D

(ng/ml)

25.5 ±10.3

20.5 ±8.6

0.095

22.1 ± 8.9

21.3 ±10.4

0.776

19.5 ± 7.4

24.4 ± 10.5

0.063

Free vitamin D

(pg/ml)

5.4 ± 2.4

4.4 ± 1.5

0.124

4.7 ±1.6

4.8 ±2.4

0.846

4.2 ± 1.5

5.3 ± 2.0

0.033

Data are expressed as mean and standard deviation (T-Test for independent samples).

Comments 7: Standardization of decimals for p-value in tables and figures would be advisable.

Response 7: Agree. We standarized to 3 decimals figures.

Comments 8: Table 5 and Figures 1 and 2 refer to the same results. It may be sufficient to show either a table or figures. Moreover, Table 5 shows statistical significance between genotypes Gc1f/Gc2 and Gc2/Gc2, while Figure 1 shows statistical significance between Gc1s/Gc2 and Gc2/Gc2. In Figure 1, the p-value of 0.0536, which is close to statistical significance, is not marked with two asterisks - information below Figure 1. A small note in the heading of Table 5: The Authors probably entered "free vitamin D libre" instead of "free vitamin D level". 

Response 8: The authors agree with this comment. Therefore, we removed Figures 1 and 2 in order not to duplicate information. We also changed “free vitamin D” and erase “libre”.

Comments 9: In the Discussion section, line 276, the word "young" could be removed if the Authors decide to change the title of the manuscript. 

Response 9: Agree. As we changed the tittle, we removed the word “young”.

Comments 10:  Standardization of the References section is necessary.

Response 10: Thank you for pointing this. We updated changes in the new version of the manuscript (pages 10-12, lines 417, 516).

Reviewer 2 Report

Comments and Suggestions for Authors

Major comments

Vitamin D is an important nutrient with more and more attention being placed on its role in health and disease. The authors of this paper attempt to quantify and analyse Vit D binding protein and free vit D in clinically stable CF patients. This is a challenging and complicated analysis to undertake as there are so many confounding factors all of which can drastically affect the interpretation of the data.

Detailed comments

Introduction:

·    - Authors need to describe in much more detail the 3 different haplotypes, their functions and how the different combinations behave in the host, what role they potentially play in disease pathogenesis.

o   Make a stranger case for why its important to understand the metabolism and homeostasis of vit D

·       -   Authors introduce the concept of the vit D SNPs in the methods section and haven’t discussed them in the introduction. These need to be introduced and explained in the introduction.

Methods:

·      - Each different methods section requires a numbered title, this was only done for half the methods.

o   Also seems as though the concomitant medication section is incomplete.

·       -  Authors need to outline more details about the study: how many centres were there, how many patients from each centre.

·         - Concomitant medications are a big confounding factor, so authors need to be clear about details, how many patients were on each medication, were patients on multiple meds at the same time? A table could be used to show this a bit better.

Results:

·       -   Authors make a point to say that only a third of patients were diagnosed through neonatal screening, why is this important? How were the remaining patients diagnosed?

·        -  It was very difficult to follow the results. Each time authors discuss a new result section, they must refer to the specific graph/table to which they are describing, otherwise its hard to follow.

·       -  Some words are still in what I assume is Spanish/Portuguese, please change to English

·       -  Authors mention several times correlation analysis or relationships, and yet don’t show any of these.

·       -  Authors state that they adjust data for a number of confounding factors, but don’t explain how they’ve done this. This is a major flaw as these factors could completely change the interpretation of the data.

o   E.g. how does one adjust for season?

·        - There are no y-axis labels on any of the graphs

o   The stats put under the graphs are not explained and don’t match what’s in the graph. Also p=0.0536 is not considered significant.

·      -   Authors mention a linear trend in graphs 3 and 4, but don’t show a linear trend and its relevant stats.

Discussion:

·        - The conclusions drawn by the authors are not convincing and are not plainly shown by their data.

·       -    More references are needed e.g line 303-305 needs some references.

·      -   How have the authors shown that their ELISA technique is more precise than the standard method (lines 351-354)?

·       -  Authors explain that significant sun exposure differences and patient adherence to vit D supplementation was not studied (line 357-360). These are glaring limitations that together with other things mentioned makes it impossible to know with certainty if their results are significant or meaningful.

Author Response

Comments 1: IntroductionAuthors need to describe in much more detail the 3 different haplotypes, their functions and how the different combinations behave in the host, what role they potentially play in disease pathogenesis. 

o   Make a stranger case for why its important to understand the metabolism and homeostasis of vit D”

Response 1: Thank you for your comment. The authors added this explanation to the introduction section in order to to expand the information included in the introduction and highlight the importance of VDBP haplotypes: “Additionally, the role of VDBP haplotypes is under study for their potential impact on diseases beyond the skeletal system. Studies conducted in recent years suggest that depending on the presence of certain haplotypes, the risk of certain diseases could increase, including oncological (breast, prostate, colorectal cancer), rheumatological (rheumatoid arthritis, spondylitis), endocrinological (diabetes mellitus, obesity, autoimmune thyroid diseases), pulmonary (chronic obstructive pulmonary disease, asthma, tuberculosis), and cardiovascular diseases (coronary artery disease)” (page 2, lines 92-98).

Comments 2: -   Authors introduce the concept of the vit D SNPs in the methods section and haven’t discussed them in the introduction. These need to be introduced and explained in the introduction.”

Response 2: The authors appreciate the reviewer's comment and have added an explanation in the introduction section (page 2, lines 83-89): “Vitamin D and its metabolites are closely related to serum proteins, with the vitamin D-binding protein (VDBP) representing the most significant of these (accounting for 85-90% of the total). Approximately 10-15% is bound to albumin, with less than 0.1% circulating freely [2,3]. VDBP is a protein produced by the liver and encoded by the GC gene (location: 4q13.3). There are several known single nucleotide polymorphisms (SNPs) in the GC gene, with rs7041 (p.Asp416Glu) and rs4588 (p.Thr420Lys) being the most investigated. Due to the allelic combination of these SNPs, three main haplotypes have been described (Gc1F, Gc1S and Gc2), which frequency varies according to race and geographical distribution. It is well established that these haplotypes determine the concentration of VDBP, its affinity for 25 OH vitamin D and, consequently, the levels of free vitamin D. [4-6]. These findings could explain the lack of correlation between total 25 OH vitamin D levels and clinical manifestations [7]”.

Comments 3: Methods:Each different methods section requires a numbered title, this was only done for half the methods. 

·       Also seems as though the concomitant medication section is incomplete.”

Response 3.1.: We agree with this comment. We have added numbered tittle in each section.

Response 3.2.: We have considered corticoesteroids (oral or inhaled, CFTR modulators/potentiators, vitamin d dose and pancreatic enzimes for this study, therefore we have changed it: 2.3.5. Concomitant medication: Pancreatic enzimes (lipase UI/kg/day), corticosteroids (oral or inhalated), CFTR potentiators and modulators (Lumacaftor/Ivacaftor (Orkambi), Tezacaftor/Ivacaftor (Symkevi)) and vitamin D (UI/day). (page 4, 166-168).  

Comments 4:-  Authors need to outline more details about the study: how many centres were there, how many patients from each centre.”

Response 4: We included patients from 4 different hospitals from Spain. Each centre contributed between 6 to 23 patients. We have added this information to the manuscript (page 3, lines 114-115).

Comments 5: “- Concomitant medications are a big confounding factor, so authors need to be clear about details, how many patients were on each medication, were patients on multiple meds at the same time? A table could be used to show this a bit better.” 

Response 5: Agree. Cystic Fibrosis patients are usually on multiple meds at the same time. All patients included in this study had pancreatic insufficiency, so all of them require pancreatic enzymes. We only included patients in a stable situation, that is why we don’t have information about antibiotics or other medications that could be needed in an accute exacerbation.

Since it is an important issue, we have created a separate table for the medication instead of including it in Table 2, as we initially had done (Table 3, page 5-6, lines 218-219):

Table 3. Patient’s concomitant medication

Medication

Value

Pancreatic enzymes (n, %)

Dosis (UI/kg/day) (mean, intercuartile range)

48 (100%)

5362.74 (3672.65 – 6925.50)

Vitamin D suplements (n, %)

48 (100%)

Corticoesteroids (n, %)

Oral corticoesteroids (n, %)

Inhalated corticoesteroids (n, %)

13 (28%)

2 (15.4%)

11 (84.6%)

CFTR modulators/potentiators (n, %)

Lumacaftor/Ivacaftor (Orkambi) (n, %)

Tezacaftor/Ivacaftor (Symkevi) (n, %)

14 (31%)

4 (28.5%)

10 (71.5%)

Comments 6:Results: -   Authors make a point to say that only a third of patients were diagnosed through neonatal screening, why is this important? How were the remaining patients diagnosed?”

Response 6: Neonatal screening is important because, due to the early detection and treatment of the disease, the prognosis is better as it is generally detected even before the onset of symptoms. Screening has been recently implemented in our country, around 10-20 years ago depending on the region. Therefore, in our series, there is a high number of patients who were not diagnosed through screening and have not been able to benefit from it: the median age at diagnosis was 8 years (interquartile range 2 - 48). We do not have data on the symptoms at diagnosis in these patients. We added the months at diagnosis to table 2 (page 5, line 218-219)

Comments 7: “-  It was very difficult to follow the results. Each time authors discuss a new result section, they must refer to the specific graph/table to which they are describing, otherwise its hard to follow.”

Response 7: The authors agree. We divided the results into different sections to facilitate understanding, indicating the corresponding table or chart in each one. We have removed Figures 1 and 2 based on the recommendation of Reviewer 1.

Comments 8: “-  Some words are still in what I assume is Spanish/Portuguese, please change to English”

Response 8: Thank you for pointing this out, we are sorry about that. We have located them and changed them to English.

 Comments 9: “-  Authors mention several times correlation analysis or relationships, and yet don’t show” any of these.

Response 9: There were other correlations observed but none of them were stadistically significant: calcidiol with vitamin D dosis (r=0.1242; p=0.4005), calcidiol with age (r=0.1078; p=0.4659), free vitamin D with vitamin D dosis (r=0.1206; p=0.4141), free vitamin D with age (r=0.1473; p=0.3178). We agree that it would be important to show them, so we added them (page 6, lines 236-239).

Comments 10: “-  Authors state that they adjust data for a number of confounding factors, but don’t explain how they’ve done this. This is a major flaw as these factors could completely change the interpretation of the data. 

o   E.g. how does one adjust for season?”

Response 10: The authors have decided to adjust for factors that, according to the literature, may influence vitamin D levels: sun exposure, the season of the year when the analysis was performed (vitamin D levels are generally lower in the winter months), the dosage of vitamin D supplementation, and the presence or absence of glucose disorders.

Comments 11:There are no y-axis labels on any of the graphs”

o   The stats put under the graphs are not explained and don’t match what’s in the graph. Also p=0.0536 is not considered significant.”

Response 11: Thank you for pointing this out. We added y-axis in Figures 3 and 4 (in the new manuscript are Figures 1 and 2) (pages 8-9, lines 283-291).

Comments 12: “-   Authors mention a linear trend in graphs 3 and 4, but don’t show a linear trend and its relevant stats.”

Response 12: Thank you for pointing this out. We mistakenly stated that it was a linear trend, but it is actually a trend test for ordered groups. We have decided to replace the Cuzick’s test with the Jonckheere-Terpstra statistical test. We have modified Figures 3 and 4 (now Figures 1 and 2 in the new manuscript), and we have made the following change: “A Jonckheere-Terpstra test showed that there was a statistically significant trend of higher median calcidiol and free vitamin D with VDBP polymorphisms ordered by VDBP affinity (from Gc2/Gc2, Gc1/Gc2 to Gc1/Gc1), TJT = 223, z = 2.189, p = 0.0264 and TJT = 215, z = 2.129, p = 0.0324, respectively” (page 8, lines 277-281).

Figure 1. Relationship between polymorphisms grouped into 3 categories ordered by VDBP affinity (Gc2/Gc2, Gc1/Gc2 to Gc1/Gc1) and Vitamin D Levels.
Please see the attachment

Figure 2. Relationship Between Polymorphisms Grouped into 3 Categories ordered by VDBP affinity (from Gc2/Gc2, Gc1/Gc2 to Gc1/Gc1) and Vitamin D Levels.
Please see the attachment

Comments 13: Discussion: “ - The conclusions drawn by the authors are not convincing and are not plainly shown by their data.”

Response 13: The Authors tried to sum up the most important findings of the study in the conclusion. We also have modified it by adding a very important part of the results: VDBP polymorphisms influence serum vitamin D and free vitamin D levels in CF patients (page 11, lines 387-388). All conclusion’s information is included in the manuscript: “In CF, as in the general population, the most frequent VDBP polymorphism in the Caucasian race is Gc1s (page 6, line 213). VDBP polymorphisms influence serum vitamin D and free vitamin D levels in CF patients (Tables 4, 5 and 6, page 6 and 7). There is a good correlation between free vitamin D and calcidiol levels (page 6, lines 227-228), indicating that the determination of the latter in CF does not seem to provide additional benefits .

Comments 14: “ -    More references are needed e.g line 303-305 needs some references.” 

Response 14: The authors have added 3 more references, including one reference in line 303-305.

Comments 15: -   How have the authors shown that their ELISA technique is more precise than the standard method (lines 351-354)?”

Response 15: The authors are referring to the fact that the determination of free vitamin D using the ELISA technique is more accurate than determining it through formulas, which is called calculated vitamin D. There are articles that compare both methods, and the results are different [*]. We add this explanation to the discussion section (Page 10, pages 366-369)

[*] Schwartz JB, Lai J, Lizaola B, Kane L, Markova S, Weyland P, Terrault NA, Stotland N, Bikle D. A comparison of measured and calculated free 25(OH) vitamin D levels in clinical populations. J Clin Endocrinol Metab. 2014 May;99(5):1631-7. doi: 10.1210/jc.2013-3874. Epub 2014 Jan 31. PMID: 24483159; PMCID: PMC4010704.

Comments 16: “-  Authors explain that significant sun exposure differences and patient adherence to vit D supplementation was not studied (line 357-360). These are glaring limitations that together with other things mentioned makes it impossible to know with certainty if their results are significant or meaningful.”

Response 16: The authors agree that not knowing exactly the level of sun exposure and adherence to vitamin D supplementation are clear limitations of the study.

However, although adherence of supplementation was not directly measured, CF patients usually receive vitamin D supplementation through multivitamin complexes that also include vitamin D,  vitamins A (retinol) and E (alfa-tocoferol), which were measured , showing normal values in all patients except in one of them, suggesting that adherence to supplementation in almost all the patients included in the study was adecuate. This data was not initially included in the manuscript but we added in Table 2. We also added this part to the discussion.

Regarding sun exposure, although it was not directly analyzed, the season of the year when the calcidiol determination was performed was considered in the analysis of the results, so it is indirectly analyzed as well. It is also worth mentioning that, although individual sun exposure may vary, all patients belong to the same race and live in the same country, so exposure is expected to be similar for all of them, considering the minimal variation in latitude between the different centers involved in the study (pages 10-11, lines 373-384).
